# Developing Cancer Quality of Care Indicators to Quantify the Impact of a Global Destabilization of the Care System (COLLAT-COVID)

**DOI:** 10.3390/cancers17101680

**Published:** 2025-05-16

**Authors:** Nathalie Piazzon, Julie Haesebaert, Philippe Michel, Anne Sophie Belmont, Vahan Kepenekian, Gery Lamblin, Charlotte Costentin, Julien Péron

**Affiliations:** 1Service de Gynécologie Hôpital de la Croix-Rousse, Hospices Civils de Lyon, 69004 Lyon, France; nathalie.piazzon@chu-lyon.fr; 2Service de Recherche et Epidémiologie Cliniques, Pôle de Santé Publique, Hospices Civils de Lyon, 69424 Lyon, France; julie.haesebaert01@chu-lyon.fr; 3Service Promotion, Prévention, Santé Populationnelle, Direction Qualité Usagers et Santé Populationnelle, Hospices Civils de Lyon, 69424 Lyon, France; philippe.michel@chu-lyon.fr; 4Plateforme Transversale de Recherche Clinique de l’Institut de Cancérologie, Hospices Civils de Lyon, 69495 Pierre-Bénite, France; anne-sophie.belmont@chu-lyon.fr; 5Service de Chirurgie Digestive et Oncologique, Hôpital Lyon Sud, Hospices Civils de Lyon, 69495 Pierre-Bénite, France; vahan.kepenekian@chu-lyon.fr; 6Service de Gynécologie Obstétrique, Hôpital Femme-Mère-Enfant, Hospices Civils de Lyon, 69677 Bron, France; gery.lamblin@chu-lyon.fr; 7Service Hépato-Gastroentérologie et Oncologie Digestive, CHU Grenoble Alpes, 38000 Grenoble, France; ccostentin@chu-grenoble.fr; 8Service d’Oncologie Médicale, Hôpital Lyon Sud, Hospices Civils de Lyon, 69495 Pierre-Bénite, France

**Keywords:** COVID-19, crisis-responsive quality indicators (QIs), cancer pathologies, health system disruptions

## Abstract

The COVID-19 pandemic severely disrupted healthcare systems, particularly affecting cancer care due to delayed diagnoses and treatments. To assess the impact of such crises, this study developed a set of hospital-based quality indicators (QIs) for four cancer types: breast cancer, hepatocellular carcinoma, gynecological cancers (excluding ovarian cancer), and peritoneal carcinomatosis. A multidisciplinary team followed a structured process, including a literature review, expert panel validation using the RAND/UCLA method, and final selection by a steering committee. Among 150 initially identified indicators, 49 were validated, with most focusing on care processes such as diagnosis, treatment, and therapeutic delays. Two indicators were common to all four cancers: multidisciplinary team discussions and psychological support consultations. This study highlights the feasibility of developing QIs tailored to health crises. The next steps will involve real-time implementation, international validation, and integration into healthcare policies to enhance crisis preparedness and ensure continuous quality improvement in cancer care.

## 1. Introduction

The COVID-19 pandemic, which began in 2020, had a significant impact on healthcare systems worldwide, leading to widespread lockdowns, travel restrictions, and reorganizations among healthcare providers to limit virus transmission and address increased pressure on health services.

In France, a series of successive and graduated national directives mandated extensive rescheduling of medical procedures across all healthcare establishments. On 16 March 2020, the government ordered the “postponement of any non-urgent surgical or medical activity, while taking into account potential risks to patient outcomes” [1]. Furthermore, during the initial lockdown, organized screening programs for breast, colorectal, and cervical cancer were suspended [2,3].

The effects on cancer care were particularly severe due to the potential negative consequences of delayed diagnoses, the complexities of cancer treatment pathways, the frequent need for high-risk surgeries and intensive care, and the heightened vulnerability of cancer patients to COVID-19 [4,5]. In early 2020, the first lockdown in France led to a dramatic decline in cancer screenings, diagnoses, and treatments, with the notable exception of chemotherapy [6,7,8,9]. In contrast, subsequent public health measures, including a second lockdown in late 2020, did not significantly impact cancer care delivery [10].

During times of healthcare system strain, as evidenced by the COVID-19 pandemic, monitoring care quality becomes crucial for informing and supporting local and national organizational initiatives. Real-time quality of care indicators were used to adapt health policies and provide operational guidance for healthcare systems [11,12]. Quality indicators assess specific healthcare processes or outcomes, and their key attributes include reliability (absence of measurement bias), validity (accurately measuring what they are intended to assess), relevance, actionability (usefulness for policymaking, monitoring, or strategy development), and feasibility [13]. A widely recognized conceptual framework for health system performance measurement, developed by the OECD, helps member countries prioritize areas for improving care quality [13].

Indicators that monitor cancer patient outcomes, such as five-year survival rates, are routinely used in many countries. While improving survival remains the ultimate goal, data on intermediate outcomes, processes, and healthcare structures are essential for guiding health system policies. A wide range of cancer quality indicators exists, covering aspects such as diagnosis, treatment, prevention, follow up, palliative care, rehabilitation, and even research [14,15]. However, in many countries, these indicators are not routinely available and require significant effort to collect, limiting their usefulness in informing real-time health system policies during crises, like the COVID-19 pandemic [16]. In this context, the development of indicators specifically adapted to disruptions in care delivery during global health crises may represent a relevant approach. The indicators developed should be designed not only as quality indicators but also as true Key Performance Indicators (KPIs), capable of guiding strategic decision-making during times of crisis. Indeed, when specifically designed to be sensitive to unstable contexts, quality indicators can serve a dual purpose: monitoring clinical practices while acting as strategic management tools for decision-makers [17,18]. These indicators, therefore, aim not only to assess the quality of oncology care but also to measure the responsiveness, resilience, and adaptability of healthcare structures. This hybrid positioning reflects an integrated approach that combines clinical relevance with organizational utility.

The primary challenge in creating cancer-specific indicators lies in the heterogeneity of cancer as a disease, with each tumor type following a distinct care pathway. This article aims to describe the process of identifying and developing a set of hospital quality of care indicators for four cancer types to monitor the impact of care reorganization during health crises.

## 2. Materials and Methods

This study was conducted in four regional hospitals: Grenoble University Hospital, Léon Bérard Cancer Center, Médipôle Lyon-Villeurbanne, and Hospices Civils de Lyon, ensuring that the impact of the epidemic on the Rhône and Isère regions was adequately represented.

In 2020, a multidisciplinary steering committee (SC) was established to oversee the project. The committee included two clinicians, a project manager, two public health specialists, and a biostatistician. The SC decided to develop quality indicators (QIs) for four patient cohorts corresponding to distinct cancer sites, each expected to exhibit significant heterogeneity in the impact of the COVID-19 crisis on patient workflows: breast cancer, hepatocellular carcinoma, gynecological cancers (excluding ovarian cancer), and peritoneal carcinosis. Ovarian cancer was excluded from the gynecological cancer cohort due to its natural history, which is characterized by frequent dissemination into the peritoneal cavity. In this study, a dedicated cohort for peritoneal carcinomatosis, regardless of the primary tumor site, was established. Therefore, including ovarian cancer in the gynecological cancer group was considered redundant. Its analysis was integrated into the peritoneal cancer cohort to ensure nosological and analytical consistency.

For each cohort, a clinical referent was appointed based on their expertise. The SC, in collaboration with the clinical referents, was responsible for assembling both the bibliography panel and the expert panel.

This study followed five key stages for each cohort as follows:

1. Literature Review: A bibliography panel, composed of two to three clinical experts supported by public health specialists and methodologists from the SC, conducted a literature review using PubMed and Google Scholar to identify relevant studies and reviews on QIs. Additional searches were conducted on official state websites and documents to gather indicators developed by national organizations recognized for promoting patient care and safety. For each identified QI, the panel recorded its title, calculation method, inclusion and exclusion criteria, and bibliographic references.

2. Selection of Indicators: Based on the indicators identified in the previous stage, the SC selected those deemed appropriate for this study. Indicators were excluded if they did not assess hospital-based quality of care, were too similar to others, or posed significant measurement challenges (e.g., involving multiple components or lacking clarity).

3. Content Validation: Content validation was conducted using the RAND/UCLA method [19,20], a modified Delphi technique involving a multidisciplinary panel of experts and anonymous scoring cycles. This method helps identify areas of agreement and disagreement among medical experts [20]. The validation process involved consensus opinions from an expert panel of 5 to 11 clinical specialists, along with a methodological expert. This study was conducted regionally, with participation from professionals across the four hospitals mentioned.

Two rounds of consensus were implemented. The first round was conducted via an electronic questionnaire sent by email. The second round took place through videoconferencing or in-person meetings. The questionnaire, created using the Mesydel platform (2021), included five closed questions per indicator, with responses measured on a 10-point Likert scale. The selection criteria assessed were (a) clinical relevance, (b) inter-institutional variability, (c) reproducibility, (d) sensitivity to change, (e) measurability within a short timeframe, and (f) suitability for assessing the impact of the COVID-19 crisis. Each indicator aimed to measure the effect of the COVID-19 crisis on care quality within the cohort.

Experts rated their agreement with the six selection criteria on a 9-point Likert scale (1 = strongly disagree, 9 = strongly agree). A criterion was validated if it achieved a median score of ≥7 and demonstrated consensus among voting members (“consensus to retain”). An indicator was considered validated when all selection criteria met the required threshold.

QIs that failed to achieve validation in the first round were reviewed in the second round. Experts were provided with a summary of their initial responses, along with anonymous group responses, to facilitate informed re-evaluation. After discussing criteria that had not reached consensus, experts were invited to re-vote on the same selection criteria.

4. Final Selection: QIs that achieved validation for clinical relevance and sensitivity to change but did not meet other criteria were submitted to the SC for a final decision. The final set of selected QIs was shared with both the bibliography panel and the expert panel for feedback.

5. Pilot Study: A pilot study for reliability analysis is currently ongoing and falls beyond the scope of this manuscript.

## 3. Results

### 3.1. Quality Indicator Selection Process

The selection of quality indicators (QIs) took place between November 2020 and June 2021. A total of 11 clinicians participated in the bibliographic panels, with representation as follows: breast cancer (*n* = 3), hepatocellular carcinoma (*n* = 3), gynecological cancer (*n* = 3), and peritoneal carcinomatosis (*n* = 2). Each panel was supported by at least one methodologist (Table 1).

The bibliographic panels initially identified 150 indicators: 80 for breast cancer, 24 for hepatocellular carcinoma, 20 for gynecological cancer, and 26 for peritoneal carcinomatosis. Following discussions, the steering committee selected 74 potential indicators and submitted them to expert groups for content validation. Although the bibliographic panels worked independently, several QIs were common across all four cancer types, particularly those related to care relevance, delays between diagnosis and therapeutic procedures, treatment modalities, and management processes (Figure 1).

A total of 23 clinical experts participated in the expert panels, distributed as follows: breast cancer (*n* = 6), hepatocellular carcinoma (*n* = 7), gynecological cancer (*n* = 6), and peritoneal carcinomatosis (*n* = 5). The expert panels validated 38 indicators after two rounds of voting. Additionally, 13 more QIs were considered valid based on clinical relevance and sensitivity to change, even though they did not meet all selection criteria. Ultimately, the steering committee selected 49 indicators (Figure 1).

### 3.2. Nature of Validated Indicators

The validated indicators covered all phases of the patient care pathway and were categorized according to the three domains of the Lancet Global Health High-Quality Health Systems framework: foundation, care process, and quality impact [21]. The foundation domain includes the facilities, personnel, and tools necessary for delivering care. The care process domain encompasses indicators related to competent, timely, and effective care, as well as patient experience. The quality impact domain reflects positive health outcomes, such as reductions in morbidity and mortality.

A significant majority of the selected QIs (92%) focused on monitoring hospital quality of care through the care process, while only 8% pertained to quality impact. Notably, no foundation indicators were validated in this study.

Despite variability in indicators by disease type, there were notable similarities in the proportions of different indicator types selected. Specifically, indicators related to the care process were particularly well represented, with proportions ranging from 82% to 100% (Figure 2).

The twelve QIs selected for the breast cancer pathway (see Appendix A, Table A1) were all process indicators: three focused on the diagnostic process, three on treatment modalities, five on delays before or between treatments, and one on staging.

The hepatocellular carcinoma pathway included eleven indicators (see Appendix A, Table A2), comprising two quality impact indicators and nine care process indicators: three focused on the diagnostic process, five on treatments, and one on delays before or between treatments.

The gynecological cancer pathway (excluding ovarian cancer) featured eight process indicators (see Appendix A, Table A3): three related to the diagnostic process, three to treatments, and two to delays before or between treatments.

The peritoneal carcinomatosis pathway included eighteen indicators (see Appendix A, Table A4), of which two were quality impact indicators and sixteen were process indicators: one focused on the diagnostic process, nine on treatments, and six on delays before or between treatments.

Notably, two indicators were common across all four pathways: the number of new cases presented at cancer multidisciplinary team meetings and the number of consultations with psychologists or psychiatrists.

These indicators collectively encompass all stages of the treatment process: diagnosis (10/49; 20.4%), treatment (36/49; 73.5%), staging (1/49; 2%), counseling (1/49; 2%), follow up (1/49; 2%), and therapeutic delays (2/49; 4.1%).

## 4. Discussion

The present study demonstrates the feasibility of developing a multidisciplinary set of QIs tailored to multiple cancer cohorts in the context of a global disruption of healthcare systems. We identified a total of forty-nine QIs across four cancer care pathways: twelve for breast cancer, eleven for hepatocellular carcinoma, eight for gynecological cancers, and eighteen for peritoneal carcinomatosis.

This approach to improving the quality and safety of care was guided by a steering committee and a dedicated team of experts. The establishment of a multidisciplinary team of clinicians allowed for the integration of diverse perspectives and strengthened collaboration among the participating hospitals. The selected indicators underwent professional consensus and align with the existing literature and best practice recommendations. The RAND/UCLA method provided a robust methodological framework for thoroughly evaluating each indicator. Through a comprehensive, structured, and evidence-based approach, we identified specific indicators relevant to the four cancers studied, covering the entire patient care pathway and facilitating coordination among various stakeholders.

Currently, these QIs are being assessed for their feasibility of implementation and their ability to reliably measure relevant outcomes.

Our study revealed a high proportion of process indicators (92%) compared to historical QI sets in oncology [15,21,22]. This finding can be attributed to our focus on indicators specifically designed to assess the effects of global healthcare disruptions, such as those experienced during the COVID-19 pandemic. Foundational indicators may have been considered less likely to be affected by such disruptions, as they primarily evaluate organizational dimensions, which fell outside the scope of this study. Quality impact indicators were selected less frequently, likely because many can only be assessed after a significant post-treatment interval, making them less effective for real-time monitoring of care quality.

Process indicators offer several advantages in this context. They can be readily extracted from patient records or other data sources (such as cohorts and registries), and some could potentially be derived from medico-administrative data [23]. These indicators are particularly useful for evaluating changes in healthcare practices [24] and identifying deficiencies in patient care over time. Importantly, only quality indicators supported by scientific evidence on process and outcome evaluation were selected. Additionally, process indicators are typically easier for healthcare providers to interpret, offering actionable insights that facilitate the replication of corrective interventions and the generation of generalizable knowledge for implementing complex healthcare improvements [25]. Beyond these general advantages, the indicators developed in this study were designed with the specific challenges of healthcare crises in mind. They focus on critical steps in the care pathway that are highly vulnerable to disruption, such as diagnostic and treatment delays. Their selection was based on feasibility for rapid data collection and real-time use, making them particularly relevant for monitoring care quality under emergency conditions [17,18]. The inclusion of a crisis-specific criterion in the RAND/UCLA validation process further ensures their contextual appropriateness, while their methodological simplicity supports future replicability and use in preparedness strategies.

The integration of both process and outcome indicators enables a comprehensive assessment of care relevance and coordination, thereby illustrating the tangible impact of the COVID-19 crisis and potential future crises. It is crucial to recognize that all indicator classifications are interconnected; thus, indicators should not be analyzed in isolation but rather within a holistic framework that considers the entire patient care pathway.

At this stage, the selected QIs have yet to be collected for the COVID-19 pandemic year and preceding years. Consequently, their reliability and feasibility for data collection remain to be evaluated. A previous study proposed breast cancer-specific QIs that were automated using the French real-life medico-administrative cancer database to develop a standardized set for breast cancer care [23]. In that study, 10 indicators were selected compared to 12 in our study, despite both being derived from similar international research [21]. Notably, only four QIs were common to both studies, highlighting the need to tailor indicator sets to their intended use and feasibility of data collection, as a standardized set may not be universally applicable across different healthcare settings.

A significant limitation of our study was the absence of patient partners in the expert panels due to the health context at the time of this study’s initiation. Including patients in this type of research is strongly recommended [25,26], as their perspectives could have fostered a more patient-centered approach. Their participation would have allowed for a better consideration of key aspects of patients’ lived experiences, such as their journey through the care pathway and the impact on their quality of life. Their absence may, therefore, have influenced the weighting or selection of certain indicators closely linked to patients’ subjective perceptions, particularly those related to treatment delays or access to supportive care services [27]. As such, the selected QIs will require validation by patient panels in future studies.

Additionally, it is important to note that all experts involved were affiliated with French institutions, and the applicability of the selected QIs may vary in other national contexts.

While this study was conducted within the French healthcare system, adapting and validating the selected indicators in non-French-speaking or resource-limited contexts represents an important area for future work. Most of the process indicators focus on core steps of the oncology care pathway, such as diagnosis, treatment, and follow up, which are broadly applicable across healthcare systems. However, contextual adaptation will be necessary, including cultural and organizational validation, assessment of data collection feasibility, and alignment with local clinical relevance. A two-phase approach could be envisioned: initial adaptation through consultation with local experts and patient representatives followed by pilot testing to evaluate feasibility, reproducibility, and sensitivity to change in diverse settings.

The pandemic has underscored the critical importance of preparedness in the healthcare sector to ensure optimal patient care and equitable access to medical services. Developing indicators specifically designed to monitor care quality during periods of healthcare disruption is a key component of crisis preparedness. These indicators must be validated, implemented, and embraced by the medical community to be effectively utilized in future crises.

The next phase of this project will focus on assessing the reliability and reproducibility of the validated indicators, as well as evaluating the feasibility of real-time data collection. Conducting this study across four regional hospitals will provide a representative overview of the pandemic’s impact. Ultimately, the goal is to expand this research into an international study, introducing standardized indicators while considering the specific characteristics and cultural contexts of each country.

## 5. Conclusions

This study demonstrates the feasibility of developing crisis-responsive QIs to monitor cancer care during health system disruptions. Future work will focus on their real-time implementation, validation in international settings, and integration into healthcare policies to enhance crisis preparedness.

## Figures and Tables

**Figure 1 cancers-17-01680-f001:**
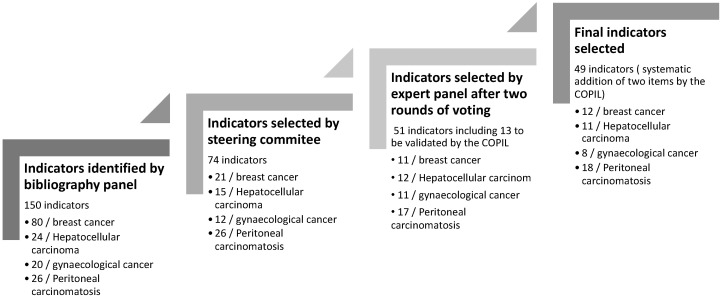
Indicator selection process.

**Figure 2 cancers-17-01680-f002:**
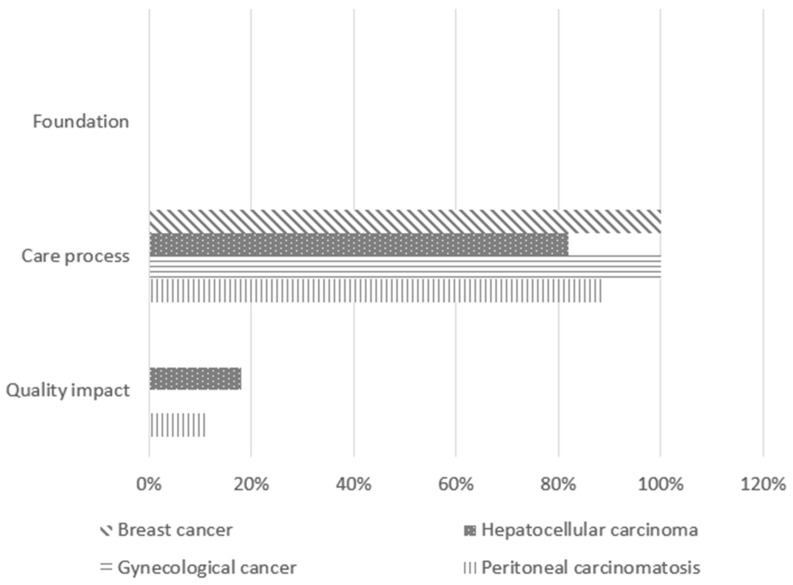
Nature of the validated quality indicators.

**Table 1 cancers-17-01680-t001:** Expert panel compositions.

	Breast Cancer	Hepatocellular Carcinomatosis	Gynecologic Cancer	Peritoneal Carcinomatosis
Bibliographic panel composition	Medical oncologist, *n* = 1 Surgeons, *n* = 2	Gastroenterologist, *n* = 3	Medical oncologist, *n* = 1 Surgeons, *n* = 2	Surgeons, *n* = 2
	Years of experience <10, *n* = 3 10–20, *n* = 0 >20, *n* = 0	Years of experience <10, *n* = 2 10–20, *n* = 1 >20, *n* = 0	Years of experience <10, *n* = 2 10–20, *n* = 1 >20, *n* = 0	Years of experience <10, *n* = 1 10–20, *n* = 0 >20, *n* = 1
Expert panel composition	Medical oncologist, *n* = 2 Surgeons, *n* = 3 Radiation therapist, *n* = 1	Medical oncologist, *n* = 1 Gastroenterologist, *n* = 6	Medical oncologist, *n* = 1 Surgeons, *n* = 3 Radiation therapist, *n* = 2	Surgeons, *n* = 5
	Years of experience <10, *n* = 0 10–20, *n* = 3 >20, *n* = 3	Years of experience <10, *n* = 0 10–20, *n* = 1 >20, *n* = 6	Years of experience <10, *n* = 1 10–20, *n* = 2 >20, *n* = 3	Years of experience <10, *n* = 1 10–20, *n* = 2 >20, *n* = 2

## Data Availability

Data are contained within the article (See Appendix A).

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
