# Peer review of "Developing Cancer Quality of Care Indicators to Quantify the Impact of a Global Destabilization of the Care System (COLLAT-COVID)"

_cancers, 2025, doi:10.3390/cancers17101680_

Round 1
Reviewer 1 Report
Comments and Suggestions for Authors
This manuscript makes a timely and methodologically rigorous effort to develop cancer-specific QIs (consider using KPIs as it is widely used at the hospital level) for healthcare system disruptions, such as those experienced during the COVID-19 pandemic. It is an important contribution to the field of oncology quality assessment, particularly in crisis contexts.
The study addresses an important gap by developing QIs that are responsive to healthcare crises. However, the authors should do a better job of explaining how these indicators are different from conventional QIs and how they add value in emergencies.
The lack of patient representatives in the expert panels is a major limitation. Although the authors mention this, they could also discuss how this omission may have impacted the selection of indicators and suggest ways of increasing patient engagement in the future.
The paper would benefit from a more detailed discussion on how the indicators could be adapted or validated in non-French contexts, particularly in low-resource settings.
The manuscript also mentions that ongoing pilot testing is currently being conducted. Including some preliminary feasibility findings or timelines for the completion of the study (if possible) would add strength to the discussion.
- P. 2, lines 27–28: Explain why ovarian cancer was taken out of the gynecological cancers and what the implications of this decision were.
- Some indicators lack denominators or clear inclusion criteria. Ensure that reporting is consistent for reproducibility (e.g., Table A4).
- Coloured figures (figure 2 at least) could be used to enhance the visual distinction between the different types of indicators.
Reviewer 2 Report
Comments and Suggestions for Authors
The study is an interesting one and could have an important practical applicability in the field of oncology through the set of quality indicators (QIs) it proposes.
It is clearly structured, the methodology is appropriate and the results are well argued.
However, in order to enhance the value of this study, I would suggest the authors to add some additional explanations:
- Why quality impact indicators were selected less frequently by experts. If possible, some short interviews with the experts, or at least with the steering committee members. The explanation given (lines 242-244) is plausible, but it would still be interesting to know the opinion of the experts or the steering committee.
- Also, I think that the lack of foundational indicators should be further argued, because even if (as the authors state) the evaluation of the organizational dimension is not the object of this study, during the COVID 19 pandemic many clinical activities (including oncology) were significantly affected by different structural shortcomings.
Finally, I wish the authors success in continuing this interesting study.
Author Response
Please see attachement
